# Aqueous Prostaglandin Eye Drop Formulations

**DOI:** 10.3390/pharmaceutics14102142

**Published:** 2022-10-09

**Authors:** Phatsawee Jansook, Thorsteinn Loftsson

**Affiliations:** 1Faculty of Pharmaceutical Sciences, Chulalongkorn University, 254 Phyathai Road, Pathumwan, Bangkok 10330, Thailand; 2Faculty of Pharmaceutical Sciences, University of Iceland, Hofsvallagata 53, IS-107 Reykjavik, Iceland

**Keywords:** ocular hypertension, prostaglandin analogues, aqueous solubility, chemical stability, drug delivery, intraocular pressure

## Abstract

Glaucoma is one of the leading causes of irreversible blindness worldwide. It is characterized by progressive optic neuropathy in association with damage to the optic nerve head and, subsequently, visual loss if it is left untreated. Among the drug classes used for the long-term treatment of open-angle glaucoma, prostaglandin analogues (PGAs) are the first-line treatment and are available as marketed eye drop formulations for intraocular pressure (IOP) reduction by increasing the trabecular and uveoscleral outflow. PGAs have low aqueous solubility and are very unstable (i.e., hydrolysis) in aqueous solutions, which may hamper their ocular bioavailability and decrease their chemical stability. Additionally, treatment with PGA in conventional eye drops is associated with adverse effects, such as conjunctival hyperemia and trichiasis. It has been a very challenging for formulation scientists to develop stable aqueous eye drop formulations that increase the PGAs’ solubility and enhance their therapeutic efficacy while simultaneously lowering their ocular side effects. Here the physiochemical properties and chemical stabilities of the commercially available PGAs are reviewed, and the compositions of their eye drop formulations are discussed. Furthermore, the novel PGA formulations for glaucoma treatment are reviewed.

## 1. Introduction

Glaucoma is a group of eye diseases that causes the progressive degeneration of the retinal ganglion cells and the retinal nerve fiber layer. The most common type of glaucoma is primary open-angle glaucoma (POAG), representing 74% of all glaucoma cases [1]. POAG is caused by the obstruction of the aqueous humor outflow within the trabecular network which increases the intraocular pressure (IOP) with consequent optic nerve damage [1,2]. Prostaglandins (PGs) are eicosanoids derived from arachidonic acid and other polyunsaturated fatty acids which have diverse biological activities, including the relaxation of smooth muscles. In general, PGs are lipophilic, chemically unstable, and poorly water-soluble compounds composed of a cyclopentane ring with two side chains [3,4]. 

In 1977, Camras, Bito and Eakins [5] showed that PGF_2α_ lowered the IOP in rabbits. It was discovered that PGs reduce the IOP by enhancing the aqueous humor outflow, and the first antiglaucoma prostaglandin analog (PGA), latanoprost, received the Food and Drug Administration’s approval between August 2000 and March 2001 [6]. Now PGAs are considered the drugs of choice for the treatment of POAG [7,8]. Currently there are five PGAs marketed as aqueous eye drops. These are 0.01% bimatoprost ophthalmic solution (Lumigan^®^, Allergan, Irvine, CA, USA), 0.005% latanoprost ophthalmic solution (Xalatan^®^, Pfizer, New York, NY, USA) and emulsion (Xelpros^®^, Sun Ophthalmics, Cranbury, NJ, USA), 0.024% latanoprostene bunod ophthalmic solution (Vyzulta^®^, Bausch & Lomb, Bridgewater, NJ, USA), 0.0015% tafluprost ophthalmic solution (Taflotan^®^, Santen, Osaka, Japan, and Zioptan^®^, Akron, Lake Forest, IL, USA/Merck, Kenilworth, NJ, USA), 0.004% travoprost ophthalmic solution (Travatan^®^, in Europe) and Travatan Z^®^ (in the USA, Novartis, Cambridge, MA, USA). All these PGA are PGF_2α_ derivatives; four are ester prodrugs of the corresponding acids, while one, bimatoprost, is an amide prodrug (Table 1). For example, latanoprost is an isopropyl ester (i.e., a prodrug) of latanoprost acid, which is a PGF_2α_ analog. Likewise, tafluprost and travoprost are isopropyl ester prodrugs of tafluprost acid and travoprost acid, respectively. Latanoprost is hydrolyzed by the corneal esterase to yield the biologically active agent latanoprost acid [6]. Bimatoprost is also rapidly hydrolyzed by ocular esterase to the biologically active bimatoprost acid [9].

Bimatoprost, latanoprost, tafluprost and travoprost appear to have very comparable efficacy regarding IOP reduction in patients with primary open-angle glaucoma [10]. Latanoprostene bunod is a prodrug of two active entities, latanoprost acid and butanediol mononitrate, which yields nitric oxide [11]. Nitric oxide lowers the IOP and improves the ocular blood flow, both of which can result in neuroprotection [12]. Thus, latanoprostene bunod might have some therapeutic advantages over the other PGAs, although the difference was shown to be insignificant with regard to the reduction in IOP [13]. An enhanced therapeutic efficacy has been obtained by combining the PGAs with non-prostaglandin IOP-lowering drugs. Examples of such combinations are 0.005% latanoprost with 0.02% netarsudil (Roclanda^®^, Aerie Pharmaceuticals, Durham, NC, USA), 0.005% latanoprost with 0.5% timolol (Xalacom^®^, Pfizer, New York, NY, USA), 0.03% bimatoprost with 0.5% timolol (Ganfort^®^, Allergan, Irvine, CA, USA), 0.004% travoprost with 0.5% timolol (DuoTrav^®^, Novartis, Basel, Switzerland) and 0.0015% tafluprost with 0.5% timolol (Taptiqom^®^, Santen, Osaka, Japan).

The monographs for latanoprost, latanoprost compounded topical solution, travoprost and travoprost ophthalmic solution are in the USP43-NF38, while the Ph. Eur. 10.3 only has a monograph for latanoprost. The following is a review of the physiochemical properties of the PGAs currently used in ophthalmology, their solubilization and stability in aqueous solutions and the composition of their eye drop formulations.

## 2. Physicochemical Properties and Eye Drop Formulations

PGAs reduce the IOP by ciliary muscle relaxation and increased aqueous humor outflow [14]. Thus, when applied topically to the eye, the PGA molecules must permeate the cornea into the aqueous humor to access their receptors. P_o/w_ is the partition coefficient (i.e., the concentration ratio at equilibrium) of an uncharged molecule between 1-octanol and water, while D_o/w_ is the partition coefficient of an ionizable compound at some fixed pH or ionization. Compounds with low P_o/w_ are hydrophilic and, in general, water-soluble, while compounds with high P_o/w_ are lipophilic and poorly soluble in water. The optimal LogP_o/w_ value (i.e., 10-logarithm of P_o/w_) for drug permeation from the aqueous tear fluid, through the cornea and into the aqueous humor is between 1 and 3, in which the drugs with LogP_o/w_ values less than 1 or greater than about 3 display a decreased ability to permeate the lipophilic cornea [15]. Prostaglandin F_2α_ and latanoprost acid are fully ionized in the tear fluid with a LogD_o/w_ value much less than unity, while their PGAs (i.e., ester prodrug analogues) have LogP_o/w_ values between 3.8 and 4.8, except bimatoprost (i.e., the amide prodrug analog) which has a LogP_o/w_ value of 2.8 (Table 1). Accordingly, bimatoprost has the optimal LogP_o/w_ value for transcorneal permeation, while the acids are too hydrophilic at a physiologic pH and the other PGAs a bit too lipophilic.

All the PGAs in Table 1 are practically insoluble in water, although bimatoprost appears to be slightly more soluble than the other PGAs in the table. The more optimal lipophilicity and slightly greater solubility increases the ability of bimatoprost to permeate from the aqueous tear fluid into the eye and can explain the slightly greater efficacy of bimatoprost compared to the other PGAs [10,13]. The PGAs are very potent drugs with low aqueous solubility which are administered topically to the eye in close to PGA saturated aqueous eye drop solutions. In other words, the dissolved PGA molecules will possess a high level of thermodynamic activity in the aqueous exterior and, thus, the molecules will have the maximum tendency to partition from the aqueous tear fluid into the lipophilic cornea [16,17]. This enhances their ability to permeate into the eye in spite of their greater than optimum LogP_o/w_ value.

The PGs are derivatives of long chain fatty acids containing a substituted cyclopentane ring which are rapidly dehydrated in aqueous solutions and known to form epimers under strong acidic and alkaline conditions [18,19,20]. Additionally, PGs and their analogs contain one or more double bonds and, thus, are sensitive towards oxidation. While PGE_2_ and related PGs are very unstable in aqueous environment, PGF_2α_ and its derivatives are, in general, less susceptible to chemical degradation. The major degradation pathways of PGAs in aqueous media are hydrolysis to form the PG acids (i.e., the active form of the PGAs), epimerization, trans isomerization and oxidation. For example, known degradation products of latanoprost in aqueous solutions are latanoprost acid, the latanoprost 15-epi diastereomer and the latanoprost 5,6-*trans* isomer, as well as oxidation products, such as the latanoprost 5-keto and 15-keto derivatives (Figure 1). Latanoprost undergoes photoinduced degradation and the highly lipophilicity drug is absorbed into plastic containers [21,22,23].

Xalatan^®^ contains 0.05 mg/mL of latanoprost in an aqueous solution of benzalkonium chloride (0.02%) as a preservative, sodium chloride for adjustment of the tonicity, a pH 6.7 phosphate buffer (sodium dihydrogen phosphate monohydrate 4.60 mg/mL and anhydrous disodium phosphate 4.74 mg/mL) and water for injection. In an unopened original package, the eye drops have a shelf-life of 36 months when stored in a refrigerator (2–8 °C) protected from light. The addition of non-ionic surfactants, such as polyoxyl 40 stearate and polyethylene glycol monostearate 25, and cyclodextrins to the aqueous eye drop media will increase the shelf-life of the latanoprost eye drops [23,24,25,26,27]. It was reported that latanoprost eye drops in the presence of 2-hydroxypropyl-β-cyclodextrin were stable at 25 °C and 60% relative humidity for at least six months, while the one containing a non-ionic surfactant remained stable for up to 24 months under the same storage conditions [23,24]. The proposed mechanism is that the interaction between the ester group of latanoprost and the complex micelle of those non-ionic surfactants results in hydrolysis being inhibited [27]. For the role of cyclodextrin, it shields the ester group of latanoprost inside the cavity, providing degradation protection [26].

The degradation profile of travoprost (Figure 2) is very similar to that of latanoprost, and in aqueous solutions, travoprost is most stable at pH 6.0 ± 0.2 [28]. Travatan^®^ contains 0.04 mg/mL of travoprost, polyquaternium-1 (0.01 mg/mL) as preservative, polyethylene glycol 40 hydrogenated castor oil (2 mg/mL) which increases the chemical stability and solubility of travoprost, boric acid, propylene glycol (7.5 mg/mL), mannitol and sodium chloride in purified water. Travatan Z^®^ contains 0.04 mg/mL of travoprost in an aqueous solution containing polyethylene glycol 40 hydrogenated castor oil, and a pH 5.7 buffer-preservative system (sofZia^®^) which is composed of boric acid, propylene glycol, sorbitol, zinc chloride and purified water [29].

The aqueous eye drops compositions of tafluprost and bimatoprost are also simple aqueous buffer solutions. Tafluprost is, like latanoprost and travoprost, an isopropyl ester with its maximum stability at pH between 5.5 and 6.7, while bimatoprost is an amide with its maximum stability between pH 6.8 and 7.8. In general, amides are more chemically stable than esters of comparable structures and studies have shown that bimatoprost eye drops are more stable than, for example, latanoprost and travoprost eye drops [31,32]. Tafluprost is the first preservative-free commercially available PGA (Zioptan^®^) containing 0.015 mg/mL of tafluprost in an aqueous solution containing polysorbate 80 as a solubilizer, glycerol, phosphate buffer and disodium edetate. Unopened cartons and foil pouches should be stored in the refrigerator (2–8 °C). After the foil pouch is opened, the unit-dose containers may be stored in the opened pouch for up to 28 days at room temperature (20–25 °C) [33].

Latanoprostene bunod is a double ester prodrug releasing two active drugs upon hydrolysis (Figure 3). Thus, one would expect that this double ester would be more chemically unstable than the other monoester prodrugs, such as latanoprost. However, the shelf-life of Vyzulta^®^ in unopened containers is similar to those of the other ester PGAs. Vyzulta^®^ contains 0.24 mg/mL of latanoprostene bunod in an aqueous solution containing polysorbate 80, glycerol, 0.2 mg/mL of benzalkonium chloride, pH 5.5 citrate buffer and disodium edetate. The shelf-life of Vyzulta^®^ unopened containers is up to 3 years at 2 to 8 °C.

Most of the commercially marketed PGA eye drops mentioned above contain benzalkonium chloride as a preservative. The concentration of benzalkonium chloride in eye drops ranges from 0.004% to 0.02% [35]. Its bactericidal activity is effective against both Gram-positive and Gram-negative bacteria, including fungi. It has been reported that benzalkonium chloride can act as a penetration enhancer, increasing the penetration of drug molecules from the surface into ocular tissues [36]. However, most in vivo studies do not support these findings [37]. To decrease the overall adverse effects of 0.03% bimatoprost, mainly conjunctival hyperemia, while maintaining the IOP-lowering effect, 0.01% bimatoprost was introduced and the concentration of benzalkonium chloride was increased from 0.005% to 0.02% [38]. However, it is well-known that the common side effects of benzalkonium chloride are conjunctival hyperemia, superficial punctate keratitis and a decrease in tear production which results in ocular discomfort and inflammation [39]. Consequently, the higher benzalkonium chloride concentration in the new bimatoprost eye drops (0.01% Lumigan, Allegan, Inc.) should result in not decreased but increased side effects, such as hyperemia, which is not the case [38]. Some studies have reported that the preserved and preservative-free PGA eye drops do not differ significantly in IOP lowering efficacy [40,41,42]. This observation might promote the marketing of novel preservative-free PGA eye drops.

## 3. Novel PGA Formulations

Cyclodextrins can solubilize and stabilize PGF_2α_ and other PGs in both aqueous solutions and solid phases [43,44,45,46,47]. Likewise, cyclodextrins are known to increase both the aqueous solubility and chemical stability of the PGAs. For example, 2-hydroxypropyl-β-cyclodextrin forms a water-soluble complex with latanoprost without decreasing the IOP-lowering effect of the drug in a rabbit model [24]. According to the investigators, the eye drop solution was stable when stored at 25 °C for at least 6 months. 2-Hydroxypropyl-β-cyclodextrin has also been shown to solubilize and stabilize tafluprost [48]. Through the formation of inclusion complexes, it has been shown that propylamino-β-cyclodextrin increases the solubilization and chemical stability of latanoprost. The in vivo ocular tolerance study in rabbits revealed that the latanoprost/propylamino-β-cyclodextrin complex eye drop formulation decreases ocular irritation when compared to the commercially available latanoprost 0.005% formulation (Xalatan^®^, Pfizer Inc., New York, NY, USA) [26]. Gonzalez et al. (2007) [49] investigated the efficacy, safety and chemical stability of a novel cyclodextrin-containing latanoprost eye drop in comparison to Xalatan^®^. The efficacy and toxicological profiles of the two latanoprost eye drops were comparable, but the eye drops containing cyclodextrin displayed improved chemical stability at 25 °C and 40 °C. Latanoprost has been formulated as an aqueous eye drop microsuspension in which the microparticles consisted of solid latanoprost/γ-cyclodextrin complexes [25]. In vitro and in vivo studies in rabbits showed that the aqueous eye drop microsuspension resulted in an almost four-fold increase in topical bioavailability and a significant enhancement in the chemical stability of the drug compared to the commercial eye drops (Xalatan^®^).

Biocompatible lipid-based nanocarriers have emerged as a potential alternative to conventional ocular drug delivery systems [50,51]. These include micro- and nanoemulsions deliver lipophilic and poorly water-soluble drugs to the ocular surface [52,53,54]. The preservative-free 0.005% latanoprost cationic emulsion (Catioprost^®^) is formulated using Novasorb^®^ technology. The cationic nanoemulsion of latanoprost was as effective as Xalatan^®^ for lowering IOP and was well tolerated by the rabbit ocular surface. Additionally, it was able to promote a healing process of the injured cornea which boosted the compliance of long-term patients [55,56,57]. When compared to Travasan^®^Z in Phase II clinical data, Catioprost^®^ decreased the IOP to the same level as Travatan^®^Z and caused less conjunctival hyperemia [58]. Ismail et al. (2020) [59] developed travoprost eye drop nanoemulsions composed of labrafac lipophile^®^ and tween 80 and studied the eye drops in vivo in rabbits. The eye drops showed prolonged IOP-lowering effects, a good level of safety and no irritation in the rabbit ocular tissues. Recently, a novel ophthalmic latanoprost 0.005% nanoemulsion was prepared and the cytotoxicity on human epithelial conjunctival cells was reported by Tau et al. (2022). It revealed that the new latanoprost nanoemulsion might cause less discomfort on the eye surface than currently available latanoprost solutions [60].

Liposomes and lipids for ocular delivery of prostaglandins have also been investigated [51,61]. Latanoprost loaded unilamellar liposomes were prepared and evaluated in eye drops. However, latanoprost could not permeate from this vehicle through the corneal epithelium, and thus it did not show any IOP reduction in vivo in rabbits. After a single subconjunctival injection, the resulting liposome formulation had an IOP-lowering effect that was sustained for up to 50 days without toxic side effects [62]. Later, the same group developed a new latanoprost-loaded egg-phosphatidylcoline liposome that resulted in the sustained delivery of latanoprost for up to 90 days and 120 days in vivo in rabbits and nonhuman primates, respectively, after a single subconjunctival injection [63,64]. Niosomes are promising ocular delivery systems formed by the self-assembly of nonionic surfactants in aqueous solutions [65,66]. A novel latanoprost niosome loaded into a poloxamer gel system had a prolonged drug release and an effective reduction in the IOP of normotensive rabbits for 3 days with no ocular irritation [67].

A thermosensitive latanoprost-loaded hydrogel composed of chitosan, gelatin and glycerol phosphate demonstrated significant IOP-lowering effects in rabbits [68]. Another thermosensitive pluronic based in situ gel for latanoprost was investigated and it was found that the optimum formulation enhanced the transcorneal permeation, resulting in a rapid decrease in IOP and a high therapeutic response compared to the reference eye drops. Furthermore, latanoprost loaded in situ gelling formulation was found to be more stable under storage conditions at 4 and 25 °C than the conventional eye drops [69]. Various types of minitablets have been developed for topical drug delivery to the eye [70]. Minitablets containing bimatoprost have been successfully developed and tested in humans (Biophta, France; www.biophta.com, accessed on 11 July 2022). After its administration to the cul de sac, the tablet transforms into an in-situ gel which releases the drug continuously for 7 days. The PGA-loaded nanotechnology platforms and other ocular biomaterials for the treatment of glaucoma are summarized in Table 2.

## 4. Conclusions

Nowadays, the only therapeutic approach to treating glaucoma is lowering the IOP. Non-invasive methods for delivering antiglaucoma drugs are preferred, especially topical administration in the form of aqueous eye drops. Five PGAs, i.e., latanoprost, bimatoprost, travoprost, tafluprost and latanoprostene bunod, are currently available as topical eye drops. Most of the PGA preparations contain preservatives that can result in side effects and local irritation. The use of non-preservative eye drop formulations has improved the ocular tolerance profile. Glaucoma patients are often on chronic therapies with multiple antiglaucoma drugs prescribed by their physicians. Fixed drug combinations (i.e., more than one active compounds in the same medication) in preservative-free eye drop formulations help to reduce the number of instillations, which leads to a reduction in the adverse reactions and an improvement the patient’s compliance. The low aqueous solubility of PGA compounds and their chemical instability in aqueous media are the main drawbacks of PGA eye drop formulation development. Thus, novel and patient-friendly PGA eye drop formulations with fewer side effects, including enhanced physical and chemical stability at room temperature, need to be developed. Among the drug delivery systems mentioned above, aqueous cyclodextrin-based nanoemulsions and in situ gel systems are potential nanocarriers for PGA formulations. These systems use biocompatible and biodegradable excipients and are suitable for large-scale production, easily applied to the site of action, and improve the ocular drug bioavailability, resulting in a greater IOP reduction compared to conventional eye drop formulations.

The development of PGA ophthalmic formulations is still challenging. The various PGAs possess different physiochemical properties which may require different formulation approaches, but in general PGAs are potent drugs that possess low aqueous solubility and poor chemical stability in aqueous-based eye drops.

## Figures and Tables

**Figure 1 pharmaceutics-14-02142-f001:**
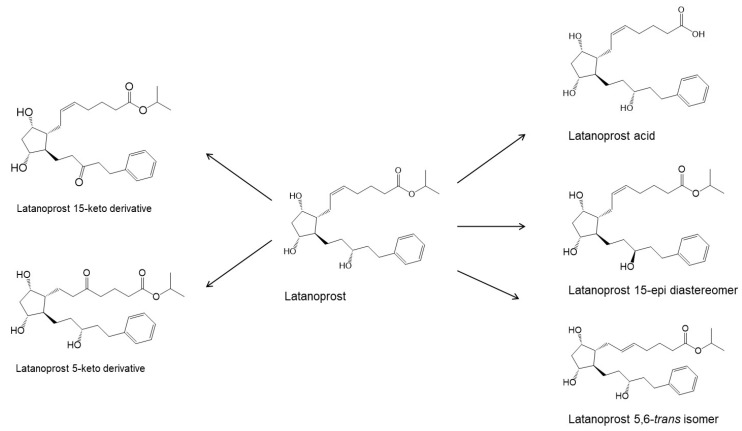
The main degradation products of latanoprost. Based on USP43-NF38 and Ph. Eur. 10.3, as well as reference [22]. Other degradation products have also been identified during forced degradation under somewhat extreme conditions [22].

**Figure 2 pharmaceutics-14-02142-f002:**
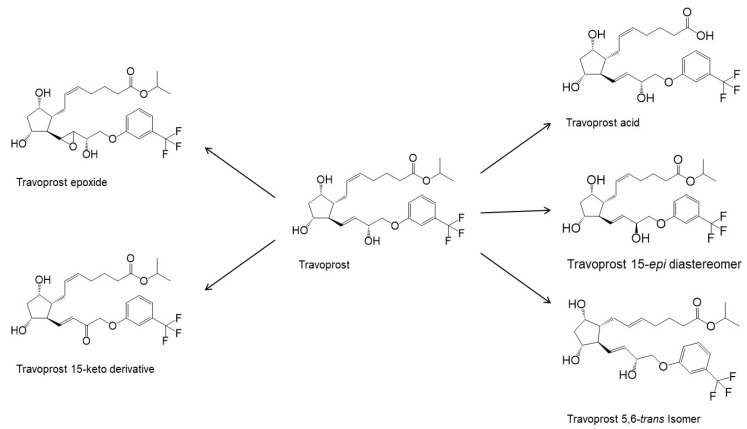
Degradation products of travoprost. Based on USP43-NF38 and reference [30].

**Figure 3 pharmaceutics-14-02142-f003:**
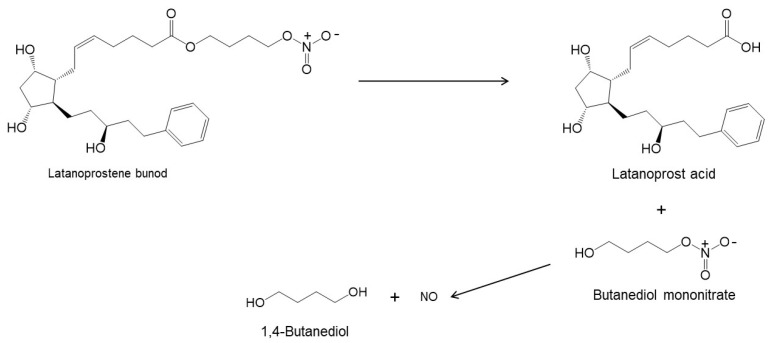
Hydrolysis of latanoprostene bunod [11,34].

**Table 1 pharmaceutics-14-02142-t001:** Structure and physicochemical properties of prostaglandin F_2α_ and its analogs, which are currently used in ophthalmology.

Prostaglandin Analog	Structure	Molecular Weight	Calculated Values ^a^	
			LogP_(o/w)_ ^b^	Solubility in Water ^c^
Prostaglandin F_2α_ (pKa 4.76)	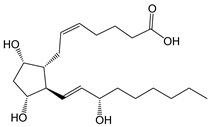	354.48	2.6 (LogD_7.0_ 0.4)	30 mg/mL (at pH 7.0)
Latanoprost acid	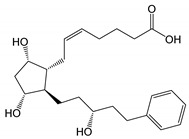	390.51	2.8 (LogD_7.0_ 0.6)	7 mg/mL (at pH 7.0)
Bimatoprost (Lumigan^®^)	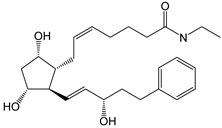	415.57	2.8	40 µg/mL
Latanoprost (Xalatan^®^, Xelpros^®^)	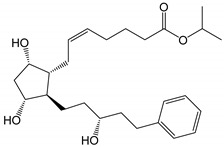	432.59	4.3	6 µg/mL
Latanoprostene bunod (Vyzulta^®^)	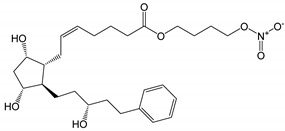	507.62	4.8	1 µg/mL
Tafluprost (Taflotan^®^, Zioptan^®^)	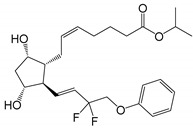	452.53	3.8	10 µg/mL
Travoprost (Travatan Z^®^)	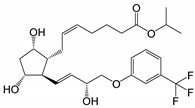	500.55	4.1	4 µg/mL

^a^ Calculated using Advanced Chemistry Development (ACD/Labs) Software V11.02 (Advanced Chemistry Development Inc., Toronto, ON, Canada). ^b^ The 10-logarithm of the calculates 1-octanol/water partition coefficient. ^c^ The calculated solubility in pure water at pH 7.0 and 25 °C.

**Table 2 pharmaceutics-14-02142-t002:** Case examples of PGA-loaded nanotechnology-based formulations and ocular biomaterial containing PGA for treatment of glaucoma.

Prostaglandin Analog	Ophthalmic Preparation	Study	Main Observation	Refs.
Nanotechnology platforms				
Latanoprost	Niosome loaded in situ gel	In vitro release and in vivo in rabbits	Prolonged drug release Effective IOP reduction in normotensive rabbits for 3 days with no ocular irritation	[67]
	Liposome	In vitro release, in vivo in rabbits and in human primate model	Sustained drug release No localized inflammatory and toxicity Not effective for IOP reduction after topical administration Successfully decreased IOP and sustained drug release after single subconjunctival injection	[62,63,64]
	Hyaluronic acid-chitosan nanoparticles	In vivo in albino rats.	A greater IOP-lowering effect in comparison to the plain drug and Xalatan^®^ eye drops	[71]
	Poly(lactic-co-glycolic acid) nanoparticles	In vitro release and in vivo in rabbits	Sustained drug release– Increasing in the drug efficacy period after applied iontophoresis Extended the period of IOP reduction for up to a week	[72]
Travoprost	DNA nanoparticles	Ex vivo in porcine cornea, in vivo in rats and mice	A long-lasting residence time on the cornea for over 60 min Enhanced the ocular drug bioavailability	[73]
	Liposome	In vitro release and in vivo in rabbits	Faster onset, longer duration and greater reduction in IOP than commercial formulation	[74]
Ocular biomaterials				
Latanoprost	Poly(lactic-co-glycolic acid) film contact lens	In vitro release and in vivo in rabbits	Initial burst release followed by sustained release Safety profile and providing a therapeutic amount of drug into aqueous humor for at least one month	[75]
	Niosome laden contact lens	In vitro release and in vivo in rabbits	Increased the drug loading capacity and prolonged drug release up to 48–96 h. Approximately three-fold enhancement in ocular bioavailability when compared to conventional contact lens	[76]
	PEGylated solid lipid nanoparticle-laden soft contact lens	In vitro release and in vivo in rabbits	Improved the drug-loading capacity and sustained drug release High drug concentrations in lower conjunctival sac compared to conventional soaked lens and drug solution	[77]
Bimatoprost	Molecular imprinted silicone contact lens	In vitro release and in vivo in rabbits	Reduced burst release and prolonged drug release Improved uptake and release kinetics of drug from the contact lens in comparison to the conventional soaking methodology	[78]
	Chitosan polymeric inserts	In vitro release and in vivo in Wistar rats	Sustained drug release IOP reduction for four weeks after application	[79]
	Ocular insert	Phase II clinical study	A greater sustained IOP-lowering effect with no adverse effects up to 6 months	[80]
Travoprost	Spanlastic nano-vesicles ocular insert	In vitro release and in vivo pharmacokinetic in rabbits	Sustained drug release Enhanced bioavailability of drug compared to marketed eye drop	[81]

## Data Availability

Not applicable.

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
