# Peer review of "Aqueous Prostaglandin Eye Drop Formulations"

_pharmaceutics, 2022, doi:10.3390/pharmaceutics14102142_

Round 1

Reviewer 1 Report

The aim of the  paper is clear.  The authors  describe the compositions of the different  PGAs preparation correctly reporting the main excipients  and their amounts. A great emphasis was attributed to the chemical aspects of the PGAs and to their physico-chemical  characteristics, such as  chemical structures and the partition coefficients.

A  minor  part concerns the presentation of the adverse affects that the use of prostaglandins  based eye drops caused on human eyes. In particular, the author  should be mention about the presence of high Benzalkonium chloride concentration  such as 0.03% in in Lumigan   (see,  doi: 10.2147/OPTH.S368214). The authors should  addressed to this  remarks and integrate the paper 

Author Response

Reviewer #1

The aim of the paper is clear.  The authors describe the compositions of the different PGAs preparation correctly reporting the main excipients and their amounts. A great emphasis was attributed to the chemical aspects of the PGAs and to their physicochemical characteristics, such as chemical structures and the partition coefficients.

A minor part concerns the presentation of the adverse effects that the use of prostaglandins-based eye drops caused on human eyes. In particular, the author should be mentioned about the presence of high Benzalkonium chloride concentration such as 0.03% in in Lumigan (see, doi: 10.2147/OPTH.S368214). The authors should be addressed to these remarks and integrate the paper 

Response:

Thank you for your suggestion. We have added:

Lines 168-183 Page 6: Most of the commercially marketed PGA eye drops mentioned above contain benzalkonium chloride as preservative. Concentration of benzalkonium chloride in eye drops ranges from 0.004% to 0.02% [35]. Its bactericidal activity is effective against both gram-positive and gram-negative bacteria, including fungi. It has been reported that benzalkonium chloride can act as a penetration enhancer increasing penetration of drug molecules from the surface into ocular tissues [36]. However, most in vivo studies do not support these findings [37]. To decrease the overall adverse effects of 0.03% bimatoprost, mainly conjunctival hyperemia, while maintaining the IOP-lowering effect, 0.01% bimatoprost was introduced where the concentration of benzalkonium chloride was increased from 0.005% to 0.02% [38]. However, it is well-known that the common side effects of benzalkonium chloride are conjunctival hyperemia, superficial punctate keratitis, and decrease tear production that result in ocular discomfort and inflammation [39]. Consequently, the higher benzalkonium chloride concentration in the new bimatoprost eye drops (0.01% Lumigan, Allegan, Inc.) should result in not decreased but increased side effects like hyperemia which is not the case [38]. Some studies have been reported that the preserved and preserved-free PGA eye drops do not differ significantly in IOP lowering efficacy [40-42]. This observation might promote marketing of novel preservative-free PGA eye drops.

Reviewer 2 Report

The paper study the physiochemical properties and chemical stabilities of the commercially available prostaglandin analogues and the composition of their eye drop formulations. Moreover novel PGA formulations for glaucoma treatment are reviewed.The paper can be accepted after minor revisions

Line 35. Please introduce the reference.

Line 40. References 5 and 6 are dated. Please insert an update.

Line 47. Please rewrite 4 in words “four”.

Table 1. Pay attention, Travoprost solubility last column, please add “l” to 4 μg/m.

Line 128. “….will increase shelf-life of latanoprost eye drops”. Please specify the shelf-life duration.

Line 165. Please write the paragraph title in bold.

Par. 3.Novel PGA Formulations. The description of the performances of the mentioned new ocular systems should be deepened. For example, lines 175-176 “decreases ocular irritation when com-175 pared to the commercially available latanoprost 0.005% formulation” data about this measurement would be more useful. The same concept for the other works mentioned.

References must be controlled such as:

Ref. 16. The title of the paper is missing.

Ref. 20. Please write the correct name of the journal

Ref. 25. The year of publication is missing

Author Response

Reviewer #2

The paper study the physiochemical properties and chemical stabilities of the commercially available prostaglandin analogues and the composition of their eye drop formulations. Moreover, novel PGA formulations for glaucoma treatment are reviewed.

The paper can be accepted after minor revisions

Line 35. Please introduce the reference.

Response:

We have added the references.

[3]        Sali, T. Prostaglandins. In Encyclopedic Reference of Immunotoxicology, Assenmacher, M., Avraham, H.K., Avraham, S., Bala, S., Barnett, J., Basketter, D., Ben-David, Y., Berek, C., Blümel, J., Bolliger, A.P., et al., Eds.; Springer Berlin Heidelberg: Berlin, Heidelberg, 2005; pp. 537-540.

[4]        Shaw, J.E.; Ramwell, P.W. Prostaglandins: a general review. Res Prostaglandins 1971, 1, 1-8.

Line 40. References 5 and 6 are dated. Please insert an update.

Response:

We have updated references.

[7]        Wang, T.; Cao, L.; Jiang, Q.; Zhang, T. Topical Medication Therapy for Glaucoma and Ocular Hypertension. Front Pharmacol 2021, 12, 749858, doi:10.3389/fphar.2021.749858.

[8]        Katsanos, A.; Riva, I.; Bozkurt, B.; Holló, G.; Quaranta, L.; Oddone, F.; Irkec, M.; Dutton, G.N.; Konstas, A.G. A new look at the safety and tolerability of prostaglandin analogue eyedrops in glaucoma and ocular hypertension. Expert Opin Drug Saf 2022, 21, 525-539, doi:10.1080/14740338.2022.1996560.

Line 47. Please rewrite 4 in words “four”.

Response:

Thank you. It was corrected.

Table 1. Pay attention, Travoprost solubility last column, please add “l” to 4 μg/m.

Response:

Thank you. It was corrected.

Line 128. “….will increase shelf-life of latanoprost eye drops”. Please specify the shelf-life duration.

Response:

We have added:

Lines 128-132 Page 4: It was reported that latanoprost eye drops in the presence of 2-hydroxypropyl-β-cyclodextrin was stable at 25°C and 60% relative humidity for at least 6 months, while the one containing non-ionic surfactant remained stable for up to 24 months under same storage conditions [23,24].

Line 165. Please write the paragraph title in bold.

Response:

Thank you. It was corrected.

Par. 3. Novel PGA Formulations. The description of the performances of the mentioned new ocular systems should be deepened. For example, lines 175-176 “decreases ocular irritation when compared to the commercially available latanoprost 0.005% formulation” data about this measurement would be more useful. The same concept for the other works mentioned.

Response:

We have added:

The in vivo ocular tolerance study in rabbits revealed that the latanoprost/propylamino-β-cyclodextrin complex eye drop formulation decreases ocular irritation when compared to the commercially available latanoprost 0.005% formulation (Xalatan®, Pfizer Inc., NY, USA).

References must be controlled such as:

Ref. 16. The title of the paper is missing.

Response:

It was appeared: Prostaglandins

Ref. 20. Please write the correct name of the journal

Response:

It was corrected:

Ochiai, A.; Iida, K.; Takabe, H.; Kawamura, E.; Sato, Y.; Kato, Y.; Ohkuma, M.; Danjo, K. Formulation design of latanoprost eye drops to improve the stability at room temperature. J. Pharm. Sci. Technol. 2010, 70, 324-332

Ref. 25. The year of publication is missing

Response:

It was corrected:

  1. Airy, R.B., J. Chiou, H. Dollinger, O. Gan, R. Jani, B. Kabra, H. Nguyen, A. Weiner. Developmental preformulation studies in the design of travoprost ophthalmic solution 0.004% (TRAVANTAN®). In Proceedings of the American Association of Pharmaceutical Scientists (AAPS) Annual Meeting, Toronto, Canada, 2002.

Reviewer 3 Report

This manuscript is a review of Aqueous Formulations of Prostaglandin Eye Drops. I think that the authors should complete section 3. New formulations of PGA. In addition to increasing the number of bibliographical references consulted

Author Response

Reviewer #3

This manuscript is a review of Aqueous Formulations of Prostaglandin Eye Drops. I think that the authors should complete section 3. New formulations of PGA. In addition to increasing the number of bibliographical references consulted

Response:

The information of nanotechnology platforms i.e., liposomes and niosomes including nanoemulsions containing PGAs were added in the main text.  Also, some other novel delivery systems of PGAs were additionally summarized in Table 2.

Reviewer 4 Report

This review is mainly about the current situation, shortage and several existing improvement methods of non-invasive delivering of prostaglandin analogues as antiglaucoma drugs. After a careful consideration of your manuscript, I don’t think it’s proper for Pharmaceutics to accept your manuscript in its status, more constructive comments and expectations should be added to the review at least.

Author Response

Reviewer #4

This review is mainly about the current situation, shortage and several existing improvement methods of non-invasive delivering of prostaglandin analogues as antiglaucoma drugs. After a careful consideration of your manuscript, I don’t think it’s proper for Pharmaceutics to accept your manuscript in its status, more constructive comments and expectations should be added to the review at least.

Response:

We have added more updated information especially in the section 3. Novel PGA Formulations. Additionally, we have added more information that may be benefit pharmaceutical formulation scientists.

Reviewer 5 Report

Excellent attempt on PGA formulations for glaucoma treatment in this article. The drug delivery systems mentioned are  aqueous cyclodextrin-based, nanoemulsions, and in situ gel systems , can authors describe some other strategies and drug delivery system which can circumvent limitations of PGA. 

Author Response

Reviewer #5

Excellent attempt on PGA formulations for glaucoma treatment in this article. The drug delivery systems mentioned are aqueous cyclodextrin-based, nanoemulsions, and in situ gel systems, can authors describe some other strategies and drug delivery system which can circumvent limitations of PGA. 

Response:

See lines 271-274 Page 9: Development of PGA ophthalmic formulations is still challenging. The various PGAs possess different physiochemical properties that may require different formulation approaches but in general PGAs are potent drugs that possess low aqueous solubility and poor chemical stability in aqueous-based eye drops.

Round 2

Reviewer 3 Report

The changes have contributed to improving the quality of the manuscript

Reviewer 4 Report

Many thanks for all of your efforts in improving your manuscript. From my piont of view, the revised article is satisfactory and can be accepted.